# From resilience to satisfaction: Defining supply chain solutions for agri-food SMEs through quality approach

Tutur Wicaksono[1,2]*, Csaba Bálint Illés[3]

1 Doctoral School of Economic and Regional Sciences, Hungarian University of Agriculture and Life Sciences, Godollo, Hungary, 2 Faculty of Economics and Business, Universitas Al Azhar Indonesia (UAI), Jakarta, Indonesia, 3 Institute of Economic Sciences, Hungarian University of Agriculture and Life Sciences, Godollo, Hungary

* Tuturwicaksono18@gmail.com, Wicaksono.Tutur@stud.uni-mate.hu, Wicaksono.Tutur@phd.uni-mate.hu

**Data Availability Statement:** All relevant data are within the manuscript and its Supporting Information files.

**Funding:** The author(s) received no specific funding for this work.

## Abstract

Since it is an important human need and many organizations are involved in the value chain, the agricultural food supply chain is exposed to various risks that arise naturally or through human actions. This study aims to develop the application of a quality function deployment approach to increase the resilience of the food supply chain by understanding customer needs and logistical risks in the food supply chain. In-depth studies with empirical analysis were conducted to determine the importance of customer needs, food supply chain risks, and actions to improve supply chain resilience of SMEs in the agri-food industry. The result shows that the top three customer needs are "attractive, bright color", "firm texture" and "fresh smell". The top three risks in the agri-food supply chain are "improper storage," "Harvest Failure" and "Human Resource Risks" and the top three resilience actions are "continuous training," "preventive maintenance," and "supply chain forecasting." The implications of this study are to propose an idea that broadens the perspective of supply chain resilience in the agri-food industry by incorporating the needs of customers in considering how to mitigate the existing risks to the satisfaction of customers, and it also highlights the relatively low skill and coordination of the workforce in agri-food supply chains.

## Introduction

Instead of using a closed strategy to compete in the industry, companies are forced to form a system to work together in today's global economic era [1]. Since food supply chain is an important human need and many organizations are involved in the value creation process, it is exposed to various risks caused by nature or human actions. Intense competition in the market in satisfying customer needs increases the demand of suppliers, which leads to increasing dependency among organizations [2]. This phenomenon makes the food supply chain vulnerable and poses the potential risk of disruption spreading throughout the supply chain [3]. Supply chain disruption can affect the company's cost efficiency, customer satisfaction, and the company's ability to generate profits [4].

**Competing interests:** The authors have declared that no competing interests exist.

To survive in the industry, companies must have a resilient supply chain. Supply chain resilience is the ability of a supply chain system to sustain the system by returning to its original state or adopting a new, more desirable state after a disruption and avoid system failure [5]. Supply chain resilience gives the firm the ability to proactively respond to changing market conditions and disruptions that may occur outside of competition. The literature used in this study focuses on operational aspects such as quality and risk in food logistics and supply chains. In addition, the academic literature related to this study provides several specific studies that can suggest a structural framework for implementing supply chain resilience.

This paper fills the research gap by providing a novelty on the application of the Quality Function Deployment (QFD) methodology in defining a supply chain resilience solution for agri-food SMEs. Agri-Food SMEs were selected as the research subject because of the importance of the supply chain and its logistical processes for a sustainable food system and its impact on food security and economic stability in general. QFD was first introduced in Japan by Akao [6] as a method for generating customer satisfaction and quality assurance points using information about customer needs. QFD is seen as a method to translate customer needs as business requirements into the company's production processes to ensure that companies understand customer expectations of the product and service development process [7].

The House of Quality planning matrix is a tool often used in QFD to link customer needs to how the firm can meet those customer needs [8]. With its development, QFD has proven to be reliable and can be applied in various industries such as automotive, hospitality, and food [9–11]. The issue of implementing QFD has also developed, which initially focused only on product design. It has since been found to be suitable for application in quality improvement, decision support systems, customer satisfaction, sustainability and also supply chain management [12–15]. This study aims to develop a QFD approach to improve supply chain resilience in agri-food SMEs by identifying customer needs, risks that affect customer satisfaction, and actions that agri-food SMEs need to take to mitigate these risks.

The objectives of this research are as follows. First, to determine the priority of customer needs. Second, to determine the priorities of risks in the agri-food supply chain. Third, to determine the priority of actions to improve supply chain resilience for agri-food SMEs by identifying customer needs, risks that affect customer satisfaction, and actions that need to be taken to mitigate these risks. We compose the research questions based on the research objectives as follows:

1. What are the priority needs of agri-food customers (CNs)?

2. What are the priority risks in the food supply chain (AFSCRs)?

3. What are the priority supply chain resilience actions (SCRAs)?

This study analyzes the relationship between these sets of variables to provide priority solutions for supply chain resilience in the food industry and provides a practical contribution for agri-food SMEs to realize a resilient supply chain system. This research is of interest to stakeholders in the sustainable food system ecosystem who are affected by food supply chain resilience. Stakeholders are generally defined as farmers, distributors (suppliers), SME retailers, food safety decision makers and end consumers. Following this introduction, Section 2 of this paper provides a literature review on quality in agri-food SMEs, food supply chain risks, supply chain resilience, application of QFD methodology in the agri-food sector and research gaps. Section 3 describes the methodology, while Section 4 presents the results and discussion. Section 5 is the final section which provides the conclusions, contributions and implications of this research.

## Literature review

### Quality in agri-food supply chain

Customer satisfaction is the main objective in planning a high-quality supply chain system [16]. Companies need to accurately identify customer needs to obtain information that can be used to design supply chains that meet customer needs in order to gain competitive advantage and increase customer satisfaction. Several studies have examined quality attributes in the context of agri-food industry [17–19]. Attributes related to product quality, such as freshness, safety, and hygiene, have always been the most important factor in agri-food product selection, followed by attributes related to price. This is because the agri-food industry has different product attributes than other industries where fresh, perishable products dominate. The product value decreases when the freshness of the product decreases due to the spoilage process [20].

In addition, service-related attributes such as responsiveness, friendliness, courtesy, convenience and knowledge are also found to play a role in the company's fulfillment of customer needs [21–23]. The pattern of classifying customer needs in the agri-food context also varies among researchers and has not yet established definitive standards. For example, Djekic et al. [24] count price, overall quality, colour, and visual freshness. Sayadi et al. [25] enumerate taste, colour, place of purchase, protected designation of origin (PDO), low acidity, organic production, environmentally friendly production, rural job creation, and rural retention.

### Agri-food supply chain risks

Risk management in agricultural supply chain is becoming a very important area of research due to the challenges associated with changing seasonality, supply and demand peaks, delivery lead time and perishability [26]. The literature on risk management in agri-food supply chain is significant, but still very limited. Dai and Liu [27] identify several risks in the agri-food business, namely the risk in the production chain, such as the impact of natural disasters leading to losses in the production process, the risk in the distribution chain, such as risks arising from transportation and unexpected loss of goods during transportation. Retail chain-based risk, which is caused by incomplete information about the product leading to the disappointment of the end consumer, and consumption chain-based risk, namely the risk arising from the estimates of customer behavior, buying habits, and customer demand based on historical data, which may be the risk of inaccurate business forecasts. In addition, Yazdani et al. [13] conducted a comprehensive analysis of agricultural supply chain risk management decision making framework which successfully identified agricultural supply chain risks including natural disasters and weather changes, biological and environmental risks, logistical and infrastructure risks, management and operational risks, public policy and institutional risks, and political and security risks, while Jianying et al. [28] stated that economic risks, such as problems related to production and logistics costs, social risks related to public policies, laws, and regulatory changes, and cooperation risks, such as lack of organizational capabilities of enterprises in terms of coordination and control caused by a weak sense of cooperation, are the most significant risks among all existing risk factors of the agricultural supply chain.

Patterson et al. [29] emphasized the importance of detailed product information related to production, processing, and preparation to reduce the risks associated with contamination of agricultural products by diseases, to increase the resilience of food systems to protect food security and learn how perishable agricultural products are handled before they reach the consumer. The researchers also pointed out the common problem of unclear information flow in the food supply chain and low awareness of the importance of understanding supply chain

risks [30]. Regulations such as mobility restrictions during the pandemic COVID-19 further increase risks in the agricultural and food supply chains. Logistical risks are associated with delivery delays due to mobility restrictions. Socio-economic risks such as the risk of child labor due to lower household income of workers producing agricultural products due to production disruptions. then environmental risks associated with the impact of agricultural product waste due to logistical delays and risks to global food security [31].

## Supply chain resilience strategies

Research related to supply chain resilience is generally concerned with the ability of supply chains to overcome the impact of unavoidable risks and return to their original operating conditions or transition to new, better conditions after a disruption [32]. Specifically, companies can take actions to improve the resilience of their supply chain by meeting unexpected customer demands. In this way, they can gain a competitive advantage by creating plans to respond to and recover quickly from supply chain disruptions [33]. From the literature of several researchers, supply chain resilience has been successfully evaluated in the context of agrifood industry using different approaches. One of them is Coopmans et al. [34], who uses mixed methods to describe supply chain resilience as a measurable assessment of a company's ability to make decisions by considering quantitative and qualitative indicators that are efficiently used to recover quickly from losses. Soni et al. [35] then present the implementation of supply chain resilience measurement using a deterministic modelling approach that is able to measure resilience with a single numerical index as a form of supply chain resilience quantification that will help companies evaluate the effectiveness of various risk mitigation strategies to facilitate supply chain resilience decision making. In addition, the qualitative discussion by Hecht et al. [36] identifies factors related to supply chain resilience in food companies including formal contingency plans, employee training, food suppliers, locations, service providers, post-event learning and infrastructure, redundant food supply, staff presence, and insurance. how these factors can provide a response in terms of risk mitigation and how these factors relate to the company's ability to withstand disruptions to ensure reliable access to food availability and safety.

Researchers have presented various ways to improve supply chain resilience. For example, Rajesh [37] stated that supply chain resilience can be improved by increasing supply chain flexibility, but to build a flexible supply chain, the complexity within a supply chain must be reduced as much as possible. Companies need to carefully develop a flexible business strategy that considers reducing elements that have less impact on supply chain resilience. A flexible supply chain has the ability to be both proactive and reactive to anticipate internal and external disruptions and uncertainties required to achieve excellent performance outcomes [38]. Li et al. [39] propose a specific example of supply chain flexibility by implementing a volume flexibility contract which can encourage producers to build capacity as the retailer's reserves and maintain channel coordination. It also helps retailers to reduce the mismatch between supply and demand as a risk sharing mechanism with lower reservation costs and more flexibility, which can encourage companies to expand and improve performance to achieve more overall profit for the supply chain as a whole. Spieske and Birkel [40] show that the visibility and collaboration aspects of supply chain planning can have an impact on increasing the sustainable resilience of the supply chain. Visibility is considered an important factor as it helps companies to obtain information about the origin of materials and components, which increases the company's understanding of its supply chain partners, which can help companies to mitigate the risk of problems at the supplier's location [41]. Collaboration in terms of interactions in the process of information dissemination, decision making, constructive communication and goal

alignment has a positive effect on increasing supply chain resilience. However, collaboration can also increase interdependence among supply chain actors, which increases the risk of creating ripple effects throughout the supply chain [42].

## Application of QFD in agri-food context

Dania et al. [43] use a combination of QFD approach, Fuzzy Analytical Network Process (FANP) and Data Envelopment Analysis (DEA) to integrate qualitative and quantitative factors in assessing the quality performance of collaboration between a sugar company and stakeholders in relation to the company's supply chain considering sustainability issues. Sayadi et al. [25] apply the QFD method to translate customer needs related to olive oil quality attributes into the practice of developing specific olive varieties that contribute most to meeting customer needs. Wicaksono et al. [20] developed a QFD model for determine priority steps to improve business quality in small and medium-sized agri-food retail companies considering the principles of open innovation. There are also some applications of the QFD approach for analyzing perceived quality in the agri-food supply chain, such as by Djekic et al. [24], who analyzed perceived quality in the supply chain of chicken products, and Djekic et al. [44] used a similar approach to study the transformation of quality perceptions of individual stakeholders in the apple supply chain. Zarei et al. [45] use QFD to identify lean enabler variables that can be practically applied to improve leanness in the food supply chain, which has an impact on reducing business costs. With certain adaptations of the House of Quality (HoQ) matrix, the QFD approach has been shown to be an effective method for organic food development [46]. Recently, the application of QFD in the agri-food sector began to lead to research related to sustainability studies [47]. There is no literature that specifically discusses the application of QFD in the context of supply chain resilience analysis in agri-food SMEs.

## Research gaps

As described above, supply chain resilience is an important factor for long-term business continuity. However, there are only two studies that specifically address the application of QFD methodology in relation to supply chain resilience in the context of the agri-food industry. Elleuch et al. [48] construct a HoQ matrix to link vulnerability factors with resilience capacity to provide a QFD-based process for a large animal feed company to increase supply chain resilience capacity. This literature only uses one HoQ matrix, does not specifically address the SME sector and customer needs variables are not considered in the HoQ matrix for supply chain resilience analysis, which are required for the supply chain quality system to meet end customer needs. Furthermore, Kumar and Kumar Singh [49] identify the level of importance of COVID19 impacts on the agri-food supply chain using Best-worst Method (BWM) and link these impacts to strategies for increasing the resilience of the agri-food supply chain using QFD.

Therefore, this study aims to fill the gap and apply the QFD approach using two HoQ matrices to prioritize supply chain resilience solutions for SME retailers in the agri-food industry by using the attribute of customer needs as a factor when considering customer satisfaction in the context of supply chain resilience, as end customers are part of the whole supply chain.

## Methodology

A model based on the QFD method is used. This method is known for providing an in-depth understanding of customer needs and then used to identify alternative solutions using the House of Quality (HoQ) matrix, which is able to define the relationship between customer needs and products (goods or services). In accordance with the objectives of this study, QFD is

an appropriate method to be used in this study because it has the advantage of translating customer needs into attributes ("how") as a form of follow-up by each functional area in meeting customer needs [8]. Originally, the QFD method was used for product development based on customer needs in a company [9,11,21], but with the development of research related to QFD, researchers have proved that QFD can be applied in a broader context, such as integration between companies in the supply chain or industry [10,15,20,45].

In this study, the QFD-based model is used to translate customer needs (CNs) into the resilience design of SMEs in agri-food retailing. The advantage of the model proposed in this study is to create a supply chain resilience design based on customer needs. Essentially, we propose an approach with two HoQ matrices. The first one links CNs to agri-food supply risk (AFSCRs) in terms of the risk performance on CNs. Then link AFSCRs to supply chain resilience actions (SCRAs) to determine how SMEs can mitigate emerging risks. The two HoQ matrix structures can be seen in Fig 1. The process used to create the HoQ matrix is described below. Preliminary consultations with industry representatives and a literature review were conducted to identify CNs and resilience designs. Resilience design in this study is an analysis of resilience measures that produce solutions in the form of priorities for action. We validated the CNs, AFSCRs, and SCRAs by conducting in-depth interviews with industry stakeholders and questionnaire surveys of end-users. Based on the data obtained from the interviews and surveys, we then developed a HoQ matrix based on the degree of importance and relationship between CNs, AFSCRs, and SCRAs. As explained earlier, there are two HoQ matrices used in the QFD approach in this study. The first matrix links CNs to agri-food supply chain risks (AFSCRs) to define AFSCRs in terms of their impact on SME business continuity based on the defined CNs attributes. CNs are represented as attributes that relate to what customers want. Therefore, SMEs need to properly identify and determine the importance of CNs, while AFSCRs are represented as factors that directly affect CNs attributes related to disruptions in achieving customer satisfaction. The initial HoQ matrix was created based on the adaptation of the QFD model by Wicaksono et al. [20] with the main steps described in detail in Section 4.1.

The second HoQ matrix will identify resilience actions that can mitigate AFSCRs identified in the first matrix. The AFSCRs will be presented as business requirements that must be carried out by SMEs in the context of risk mitigation, while the SCRAs will be presented as practical solution measures to be carried out by SMEs to mitigate the risks. As shown in Fig 1, the value assessed in the first HoQ matrix for the importance of agri-food supply chain risks becomes the starting point for building the second HoQ. Thus, the steps used to build the first

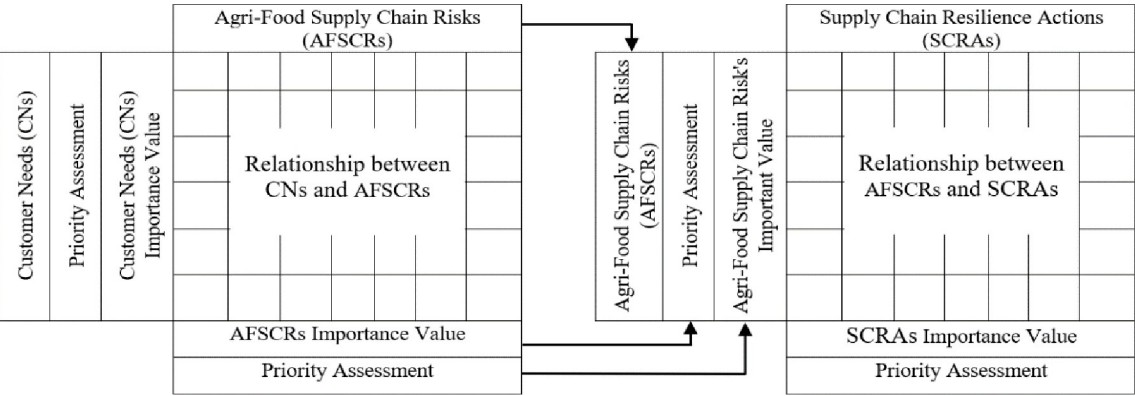

**Fig 1. First and second house of quality matrix structure of agri-food supply chain resilience.**

HoQ matrix can also be used to build the second HoQ matrix, which we will show in detail in Section 4.2.

The empirical analysis of this study was conducted among SMEs and their supply chain actors in *Kramat Jati* Central Market, East Jakarta, the largest traditional market for agricultural products in Southeast Asia, so the results can represent the agricultural products supply chain worldwide. In this study, semi-structured in-depth interviews were conducted with ten loyal customers of SME retailers to identify CNs. Similar in-depth interviews were also conducted with twenty of the largest and most influential actors in the agri-food industry, including SME owners, suppliers, and logistics actors, to identify risks in the agri-food supply chain, assess the relationship between CNs and AFSCRs and between AFSCRs and SCRAs, and assess the priority of agri-food supply chain resilience actions (SCRAs). Conducting interviews with economic actors and industry stakeholders is a common method for data collection in studies related to QFD [50,51]. This study also uses a Likert scale (1–5) questionnaire to survey 341 customers of small and medium enterprises in the agri-food industry to create a summative score to measure a construct. The customer profile can be found in Table 1. Referring to Roscoe's rule of thumb, which we quote from Sekaran and Bougie [52] which states that a sample size between more than 30 and less than 500 respondents is appropriate for most studies.

Validity and reliability tests for the items of the CNs attribute were also conducted using SPSS 25. The results of the validity test showed that each item was valid with a significance level of 0.01. In relation to the reliability test using Cronbach's alpha, it shows a value of 0.812, which means that the data are reliable to use [53]. The data collection lasted for four months, starting at the beginning of March 2021 and ending at the end of June 2021. During the data collection, the researchers always adhered to the health protocols prescribed by the local government, such as hand washing, wearing masks, and social distancing. Respondents gave verbal consent to participate in the study and none of the respondents were minors. This study was reviewed and approved by an institutional review board (ethics committee) of the Doctoral School Hungarian University of Agriculture and Life Sciences before the study commenced.

**Table 1. Customer needs (CNs) respondent profile analysis.**

|  |  | N | Percentage (%) |
|---|---|---|---|
| Age | 19–30 | 62 | 18.2 |
|  | 31–40 | 109 | 32 |
|  | 41–50 | 111 | 32.5 |
|  | >50 | 59 | 17.3 |
| Gender | Male | 124 | 36.4 |
|  | Female | 217 | 63.6 |
| Education | Higher Education/University | 140 | 41.1 |
|  | Non-higher education | 201 | 58.9 |
| Purchasing experience | 5 years or less | 53 | 15.5 |
|  | 5–10 years | 80 | 23.4 |
|  | 10–20 years | 142 | 41.8 |
|  | >20 years | 66 | 19.3 |
| Purchasing frequency | More than 2 times a week | 147 | 43.1 |
|  | Once a week as minimum | 121 | 35.5 |
|  | Once a month as minimum. | 73 | 21.4 |

## Result and discussion

### First HoQ matrix (linking customer needs and agri-food supply chain risks)

**Step 1. Identification of customer needs (CNs).** In this step, CNs are identified through interviews with stakeholders in the agri-food supply chain and literature review. Ten CNs attributes were identified and validated as shown in Table 2.

**Step 2. CNs importance value priority assessment.** The CNs are ranked according to the level of importance value and customer perception from the customer survey (Table 3). "Attractive bright color", "firm texture" and "fresh smell" are the three CNs that have high importance value and are considered most important. Customers are not too concerned about price and the possibility of contamination that may occur in agri-food products. This is confirmed by the retailers that the factors related to price and possibility of product contamination are not important for the customers in consuming agri-food product. The importance of CNs is rated on a scale of 1–5, based on a 5-point Likert scale. Scale 1 = very unimportant and 5 = very important. The importance level of customer needs is calculated using the following equation:

$$CNIV_j = \frac{\sum_{n=1}^{m} CNs_i}{m} = \frac{CN_1 + CN_2 + CN_3 \ldots \ldots + CN_m}{m} \qquad (1)$$

$CNIV_j$: is the average value of the CNs importance scores of each CNs attribute

$CNs_i$: individual CNs importance value collected from each respondent for each CNs attribute, where $i$ varies from 1 to m, where $i$ is the number of respondents, which is 341.

**Step 3. Defining agri-food supply chain risks.** This step uses a literature review and initial in-depth interviews with industry representatives to identify the technical requirements associated with supply chain risks in the agri-food industry based on their impact on meeting each of the CNs previously defined. Eight risks identified and reviewed by relevant industry representatives are listed in Table 4.

**Step 4. Relationship matrix.** Assessment of each attribute of customer needs and agri-food industry Supply Chain Risks to determine the extent of the relationship between the attributes. Constructed from averages of the results of in-depth interviews with industry players. Assessed using quantified relationship symbols representing three levels of relationship:

**Table 2. Customer needs (CNs) list.**

| Customer Needs (CNs) | Description | Reference |
|---|---|---|
| Attractive bright color | Ability to deliver products that have a color that is not pale, bright, and attractive. | [54] |
| Fresh smell | Ability to deliver products that have a natural, fresh smell that does not smell bad | [54] |
| Firm texture | Ability to deliver products that have a firm texture that is not mushy | [55] |
| Stock availability | Ability to maintain availability of products in stock | [56] |
| Cleanliness | Ability to deliver products in clean condition. | [57] |
| Tastiness | Ability to deliver products that have a good taste. | [55] |
| Price stability | Prices must remain stable, no sudden significant price increases. | [56] |
| Product variation | Ability to supply a variety of products. | [58] |
| Proper shape | Ability to deliver products that have standard shapes and no shape defects. | [59] |
| Contamination-free | Ensure that there is no contamination between different types of fresh products that affect food safety | [60] |

**Table 3. Importance value and priority ranking of the customer needs (CNs).**

| Customer Needs (CNs) | Importance Value | Priority Ranking |
|---|---|---|
| Attractive bright color | 4.61 | 3 |
| Fresh smell | 4.65 | 1 |
| Firm texture | 4.60 | 4 |
| Stock availability | 4.63 | 2 |
| Cleanliness | 4.55 | 5 |
| Tastiness | 4.43 | 7 |
| Price stability | 4.50 | 6 |
| Product variation | 3.89 | 10 |
| Proper shape | 4.24 | 9 |
| Contamination-free | 4.36 | 8 |

strong, moderate and weak. Strong relationship symbol is weighted as 9, moderate relationship symbol is weighted as 3 and weak relationship symbol is weighted as 1 (Table 5).

**Step 5. Agri-food supply chain risk priority assessment.** In this step, we determine the importance value to assess the priority order of each risk of the agri-food supply chain based on the value of the relationship matrix and the importance value of each customer need itself by summing the multiplication values of the importance value of each customer need with the value of the relationship matrix between the customer needs and the attributes of the agri-food supply chain that are connected on the same path of the matrix.

*Attractive bright color.* As the results show, attractive bright color is most influenced by the risk attributes of harvest failure and improper storage. Factors leading to harvest failure, such as pest infestation and poor storage conditions, also have a strong influence on the decline or poor quality of the product, which is reflected in a dull appearance of the external color of the product.

*Fresh smell.* Similar to firm texture, the "fresh smell" attribute is also affected by most of the potential risks in the list. For the fresh smell attribute, improper storage is also the greatest risk for affecting the fresh smell attribute. This is understandable as [Bhat and Reddy, 67] found that improper storage conditions contribute to the rapid growth of fungi in fresh agricultural produce, especially in areas of high humidity and temperature such as the tropics. The development of mold contributes to the appearance of unpleasant odor in fresh produce.

**Table 4. Agri-food supply chain risks list.**

| Potential risks | Description | Reference |
|---|---|---|
| Harvest failure | Risk of drastic loss or reduction of crops and livestock due to pest infestation in crops, disease infestation in livestock and natural disasters | [26] |
| Bullwhip effect | Inaccurate demand forecasts from retailers as they are highly responsive to demand and reinforce expectations in the surrounding supply chain. | [61] |
| Transportation accident | Risks associated with a possible accident involving a product-supporting vehicle, such as a truck accident or a sinking ship | [62] |
| Equipment failure | Failure of production support equipment to function properly | [61] |
| Improper storage | Problems related to storage during distribution, such as insufficient capacity, unsuitable temperature, poor packaging material and contamination | [63] |
| Human resource risk | Lack of skills, knowledge, concern, coordination and motivation of industry actors | [64] |
| Traffic congestion | The risk of travel time being longer than it should be due to congestion caused by queues of vehicles exceeding road capacity. | [65] |
| Criminal activities | Risk related to robbery and theft in the shipping process | [66] |

**Table 5. Quantified relationship symbols.**

| Symbols | Relationship Level | Value |
|---------|--------------------|-------|
| ● | Strong | 9 |
| ○ | Moderate | 3 |
| △ | Weak | 1 |

*Firm texture*. As shown in Table 6, Firm texture can be affected by seven of the eight potential risks, with improper storage posing the greatest risk affecting firm texture. As shown by [Park et al., 68], perishable foods are sensitive to things like storage time and temperature, and improper storage will quickly damage the texture of the product due to spoilage.

*Stock availability*. The results show that all risk attributes can pose potential threats and disruptions to stock availability, and all risks are also considered to be strongly related to these customer needs. In the context of the supply chain, stock availability is a very important factor because if the available stock is lower than the demand, it negatively affects the performance of the whole supply chain and leads to a decrease in customer satisfaction; on the other hand, overstocking leads to an increase in inventory levels and poses the risk of quality deterioration for perishable products that are sensitive to storage time.

*Cleanliness*. As customer behavior changes and hygienic aspects such as cleanliness of purchased food become more important to avoid disease outbreaks during the Covid19 pandemic, industry players can also use this as a competitive advantage by immediately improving the cleanliness aspects of food products [Han et al., 69]. The results show that personnel risk is the most influential risk for product cleanliness. In this case, the lack of specific training for industry actors on professionalism in satisfying customers is related to product quality, specifically how the process of producing clean and dirt-free products affects competitiveness and disrupts supply chains.

*Tastiness*. As the results show, the attribute "tastiness" is most affected by improper storage and equipment deficiencies. Processing equipment in direct contact with the agricultural product, as well as the shape, design and condition of packaging and storage of a product can affect the tastiness attribute based on the perception of the quality of the product.

*Price stability*. This attribute is most affected by the risk of harvest failure. The impact of harvest failure can lead to product shortages that disrupt supplies to customers. Product demand that cannot be met due to lack of product availability in the market leads to price inflation.

*Product variation*. Although the results show that product variation is not a major need of customers, the availability of a variety of product variations in the agri-food sector has a positive impact on supply chain performance and customer satisfaction. There are five types of risks on the list that can pose a threat to product variation, but only two risks that have a strong relationship, namely the risk of harvest failure, which offers the potential for disruption to the availability of product variation, and the bullwhip effect on the impact of retailers' errors in predicting the supply of customer demand for product variation.

*Proper shape*. The appearance of an appropriate product form also plays a role in customers' perception of product quality. Proper shape attribute is most influenced by human resource risk, although the relationship is moderate. With sufficient knowledge of the personnel regarding the appearance of the standard form of any product desired by the customer, the product quality control can be fulfilled.

*Contamination free*. As the results show, the attribute "contamination-free" is mainly influenced by the effects of improper storage. Improper storage techniques and temperatures, such as placing several different agricultural foods together, can lead to contamination with bacteria

**Table 6. First HoQ matrix (linking customer needs and agri-food supply chain risks).**

| Customer Needs (CNs) | Priority ranking | Importance value | Agri-food Supply Chain Risks (AFSCRs) | | | | | | | |
|---|---|---|---|---|---|---|---|---|---|---|
| | | | Harvest failure | Bullwhip effect | Transportation accident | Equipment failure | improper storage | Human resource risk | Traffic congestion | Criminal activity |
| Attractive bright color | 3 | 4.61 | △ | △ | △ | △ | ● | ○ | △ | |
| Fresh smell | 1 | 4.65 | △ | △ | △ | △ | ● | ○ | △ | |
| Firm texture | 4 | 4.60 | ● | △ | △ | △ | ● | ○ | △ | |
| Stock availability | 2 | 4.63 | ● | ● | ● | ● | ● | ● | ● | ● |
| Cleanliness | 5 | 4.55 | | | | △ | △ | ● | | |
| Tastiness | 7 | 4.43 | ● | | △ | ● | ● | ○ | | |
| Price stability | 6 | 4.50 | | | | △ | △ | △ | △ | |
| Product variation | 10 | 3.89 | ● | ● | | △ | △ | ○ | | |
| Proper shape | 9 | 4.24 | △ | △ | △ | | | ○ | △ | △ |
| Contamination-free | 8 | 4.36 | △ | | | | ● | | ○ | ● |
| Importance value | | | 175.8 | 94.8 | 64.2 | 108.3 | 258.5 | 166.4 | 77.4 | 45.9 |
| Priority ranking | | | 2 | 5 | 7 | 4 | 1 | 3 | 6 | 8 |

and other harmful microorganisms, as well as accelerate chemical and enzymatic reactions that affect the quality of agricultural foods due to improper storage practices, potentially reducing the shelf life of the product [70].

The results show that risk related to internal factors has the greatest impact. This implies that SMEs are more focused on the environmental risks of their respective organizations and less concerned about the risk factors for external disturbances related to the extensive network of the agri-food supply chain. This shows the weak coordination between organizations within the supply chain.

## Second HoQ matrix (linking supply chain risks and resilience actions)

**Step 1. Identification of risks in the agri-food supply chain.**   In this step, the agri-food supply chain risk attributes are transferred directly from the first HoQ matrix to the left side of the second HoQ matrix (Fig 1).

**Step 2. Agri-food supply chain risk priority assessment.**   Importance value and priority rank of the agri-food supply chain risk determined in the fifth step of the first HoQ matrix are directly transferred to the left side of the second HoQ matrix (Fig 1).

**Step 3. Identification of supply chain resilience actions.**   Based on in-depth interviews with industry actors and a literature review, six actions to reduce supply chain risks for agri-food SMEs were identified (Table 7).

**Step 4. Relationship matrix.**   The relationship matrix created by averaging the results of in-depth interviews with industry stakeholders and literature reviews on the relationship between food supply chain risks and supply chain resilience actions. Assessed using quantified relationship symbols representing three levels of relationships, similar to those used in the first HoQ matrix.

**Step 5. Resilience actions priority assessment.**   Determine the importance value for assessing the priority order of supply chain resilience actions by adding the multiplication values of the importance value of each agri-food supply risk and the value of the relationship matrix of the second HoQ matrix, using a similar calculation method as in step 5 in the first HoQ matrix as shown in Table 8.

*Harvest failure*. The disaster mitigation plan is the most effective solution to mitigate the risks of harvest failure due to natural disasters (force majeure). Planting annual trees with

Table 7.  Supply chain resilience actions list.

| Proposed resilience actions | Description | Reference |
|---|---|---|
| Disaster Prevention Plan | A planned program that is periodically established and evaluated to determine actions to mitigate disaster risk. | Nguyen et al. [71] |
| Preventive Maintenance | Regular and scheduled maintenance of equipment and facilities to avoid downtime due to unexpected equipment malfunctions. | Yang et al. [72] |
| Continuous Training | Continuous training programs to help personnel (industry players) improve the skills and knowledge required to minimize vulnerabilities and avoid dangerous repeat errors. | Mithun Ali et al. [73] |
| Supply chain coordination | Engagement collaboration between companies or stakeholders in a supply chain in sharing resources and information to achieve common goals with a focus on customer satisfaction. | Kim and Chai [74] |
| Forecasting supply chain | Accurately predict future patterns in supply, demand, and price of products in the supply chain by collecting data on past patterns in the supply chain and data from suppliers. | Huber et al. [75] |
| IT Utilization | The use of technology involving devices or computer systems related to software, applications, storage, and networks to effectively and efficiently manage required information. | Lezoche et al. [76] |

**Table 8. Second HoQ matrix (linking supply chain risks and resilience actions).**

| | | Priority ranking | Importance value | Supply chain resilience actions (SCRAs) | | | | | | |
| --- | --- | --- | --- | --- | --- | --- | --- | --- | --- | --- |
| | | | | Disaster Prevention Plan | Preventive Maintenance | Continuous Training | Supply chain coordination | Supply chain Forecasting | IT Utilization | |
| Agri-food Supply Chain Risks (AFSCRs) | Harvest failure | 2 | 175.8 | ● | | △ | ○ | ○ | ● | |
| | Bullwhip effect | 5 | 94.8 | | △ | ● | ● | ● | △ | |
| | Transportation accident | 7 | 64.2 | ● | ● | ● | | | △ | |
| | Equipment failure | 4 | 108.3 | ○ | ● | △ | | | △ | |
| | Improper storage | 1 | 258.5 | | ● | ● | ○ | ● | △ | |
| | Human resource risk | 3 | 166.4 | △ | ○ | ● | ○ | △ | △ | |
| | Traffic congestion | 6 | 77.4 | △ | △ | △ | ○ | △ | ● | |
| | Criminal activities | 8 | 45.9 | ● | | | ● | | △ | |
| | | | | 3141.8 | 4550.4 | 5616.6 | 2949 | 3599.3 | 3016.9 | Importance value |
| | | | | 4 | 2 | 1 | 6 | 3 | 5 | Priority ranking |

agricultural value in water-prone areas can prevent disasters such as floods and landslides. To reduce the risk of harvest failure, it is also necessary to implement a cropping plan adapted to the local climate based on the planting calendar, use better pest-resistant varieties, vaccinate livestock, and regularly monitor livestock health and environmental conditions. Training in quality management correlates least with the harvest failure attribute, as the phenomenon of force majeure is very difficult to predict even with advanced training.

*Bullwhip effect*. Accurate supply chain forecasting is an effective way to prevent the negative effects of the bullwhip effect. Such forecasting capabilities require adequate knowledge of supply chain resilience, which can be improved through continuous training. We support the findings of Mithun Ali et al. (73) that continuous training for relevant personnel, especially in the retail sector, can improve the ability to accurately forecast customer demand.

*Transportation accident*. We support the research findings of Saleh et al. [77] that preventive maintenance is the most effective measure to prevent the occurrence of accidents by regularly and routinely checking and repairing the performance condition of the transport vehicle. Training is also beneficial to improve the driver's ability to minimize driving behaviors that increase the risk of accidents, such as not driving when drowsy, not exceeding the speed limit, and other technical behaviors while driving. contingency plans can evaluate accident-prone routes and then look for new routes that can be travelled with similar or more efficient travel costs.

*Equipment failure*. When the main equipment directly related to the production process fails or is completely damaged, it takes a long time to repair, which in turn disrupts the supply chain flow. Routine and regular preventive maintenance is an important solution to prevent the occurrence of technical downtime due to unexpected failures, so that production can run on time or at least with as little technical downtime as possible [77].

*Improper storage*. Again, preventive maintenance is the most effective measure to avoid the risks posed by poor agricultural storage conditions. Regular inspections can monitor the adequacy of temperature and physical conditions of storage and overall product packaging to

determine if repairs or replacement of defective parts are needed. Accurate demand forecasting is also beneficial to minimize the risk of massive deterioration in product quality due to inventory accumulation from improper storage.

*Human resources risk*. This aspect involves workers from the agricultural sector and logistics. There is a need to develop a workforce that is multi-skilled. Therefore, there is need for continuous training of relevant players in the industry to ensure that they have the necessary skills to mitigate the risk of high customer satisfaction.

*Traffic congestion*. Using IT can effectively reduce the risk of getting stuck in traffic jams. The use of software or online mapping applications that are able to provide real-time information on road density and alternative routes with shorter travel times, as well as the best departure times, to avoid the risk of congestion. This finding is supported by the findings of Zafar and Haq's [78] research that the traffic map application in real time is able to predict the estimated arrival time of various special weather features, special conditions, holidays, and alternative road options. Then, the results of the traffic analysis are classified into the five highest traffic volume levels in the area or road to be traversed, with the highest prediction accuracy reaching 92%.

*Criminal activities*. A disaster prevention plan is also an effective solution, this time as a crime prevention measure by reviewing the route and delivery times to avoid always using the same route at the same time and avoiding certain routes at certain times that are prone to criminal acts such as robberies, by having a delivery strategy of travelling in convoy to protect each other, and by informing the police in certain areas to increase security in areas affected by crime.

Agri-food SMEs need to take immediate action such as continuous training, preventive maintenance and supply chain forecasting, which are the three main solutions that need to be implemented as an effective strategy to mitigate existing risks. The results show that the skills and knowledge of the agri-food SMEs actors in the studied supply chain are relatively low. This also affects the weak coordination among organizations in the supply chain. Efforts to improve supply chain resilience need to be carried out with systematic coordinated actions between organizations in the supply chain. Companies need to understand the importance of maximizing the benefits of the entire supply chain and not just focusing on their own profit motive. Retailers can undertake coordination and collaboration initiatives that are flexible and with guaranteed mutual benefits in terms of increased profits by minimizing costs and risks. Large computer companies such as Dell Computer have created mature supply chain integration with their "just-in-time" strategy that emphasizes real-time coordination and visibility with their suppliers. This practice can also be applied to the agri-food supply chain, where the quality of goods is highly dependent on time.

## Conclusions

Agri-food SMEs are aware of the importance of building a resilient inter-company supply chain to face intense competition in a global era complicated by the current pandemic, changing customer needs and the risks faced by industry players to meet customer needs. This research has succeeded in proposing an idea that broadens the perspective on supply chain resilience in the agri-food industry by incorporating attributes of customer needs in considering how existing risks can be mitigated to satisfy customers.

An empirical study was conducted using the QFD approach to examine the relationship between different attributes in the agri-food supply chain in Indonesia. The main findings are presented below:

1. Fresh smell, stock availability dan attractive bright color are the top three customer needs.

2. The top three risks are improper storage, harvest failure dan human resources risk.

3. Continuous training, preventive maintenance and supply chain forecasting are the top three resilience measures.

## Implications

Several theoretical implications arise from this result. First, this study is able to provide new ideas and successfully fill the literature gap presented in Section 2 by considering attributes of customer needs and combining them with risk attributes to find resilience solutions that can also satisfy customers in the agri-food supply chain of SMEs. Secondly, this study is able to provide ideas for future researchers as a reference for the practical procedural steps in using two levels of HoQ as a tool to determine the what and how in the QFD method. The thorough understanding gained from a recent literature review on agri-food quality, agri-food risk and supply chain resilience provides the basis for identifying and developing attributes for customer needs, agri-food risk and resilience to build HoQ, which is also confirmed by customers and relevant industry stakeholders. Third, this study proposes an idea for a list of customer needs, risks and resilience solutions that could become a standard for future research.

This study also has practical implications for agri-food SMEs to build a resilient agri-food supply chain. Firstly, this study has succeeded in highlighting the low skills and knowledge of agri-food actors, especially in the area of supply chain and total quality, so as to raise enthusiasm for improving skills. In addition, agri-food SMEs are weak in coordination among organizations in the supply chain as they are only concerned about internal operational benefits. In view of the findings of this study, it is recommended that agri-food SMEs, starting from producers to suppliers to distributors and retailers, can improve coordination and collaboration to overcome customer satisfaction risks.

In addition, important implications for management also emerge from this study. This industry is a labor-intensive industry. The results show that the most important risks that can negatively affect customer satisfaction are related to the lack of skills and knowledge of employees. Therefore, business owners or top managers should pay more attention to employees' skills and knowledge through continuous training.

## Limitation and future research

This study has limitations in terms of sample size as the sample survey period coincides with the period of implementation of special social distancing regulations related to the pandemic COVID -19. The number of SMEs and existing clients is less than under normal circumstances. There is still much room for research on supply chain resilience in the agri-food industry, for example, the phenomenon of vulnerability due to the bullwhip effect in the supply chain or the introduction of information technology and its application may also be explored in the future.

## Supporting information

**S1 File. Tabulation.**
(XLSX)

**S2 File. In-depth interview result.**
(DOCX)

**S3 File. Questionnaire.**
(DOCX)

## Author Contributions

**Conceptualization:** Tutur Wicaksono, Csaba Bálint Illés.

**Data curation:** Tutur Wicaksono.

**Formal analysis:** Tutur Wicaksono.

**Investigation:** Tutur Wicaksono.

**Methodology:** Tutur Wicaksono.

**Project administration:** Csaba Bálint Illés.

**Resources:** Tutur Wicaksono.

**Software:** Tutur Wicaksono.

**Supervision:** Csaba Bálint Illés.

**Validation:** Tutur Wicaksono, Csaba Bálint Illés.

**Visualization:** Tutur Wicaksono, Csaba Bálint Illés.

**Writing – original draft:** Tutur Wicaksono.

**Writing – review & editing:** Tutur Wicaksono, Csaba Bálint Illés.

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
