## [Decision Letter · Decision Letter 0]

20 Nov 2021

PONE-D-21-28048From resilience to satisfaction: defining supply chain solutions for agri-food SMEs through quality approachPLOS ONE

Dear Dr. Wicaksono,

Thank you for submitting your manuscript to PLOS ONE. After careful consideration, we feel that it has merit but does not fully meet PLOS ONE’s publication criteria as it currently stands. Therefore, we invite you to submit a revised version of the manuscript that addresses the points raised during the review process. Please consider all points raised by reviewers, with particular attention to the sample size considerations made by reviewer 2 and the overall organization of the paper made by reviewer 1 (I would recommend too a clear distinction of the types of implications). Please also note that both reviewers have raised issues about the quality of the English text and the need for a proofreading.

We look forward to receiving your revised manuscript.

Kind regards,

Claudio Soregaroli

Academic Editor

PLOS ONE

Journal Requirements:

Reviewers' comments:

Reviewer's Responses to Questions

**Comments to the Author**

1. Is the manuscript technically sound, and do the data support the conclusions?

Reviewer #1: Yes

Reviewer #2: Yes

2. Has the statistical analysis been performed appropriately and rigorously? 

Reviewer #1: N/A

Reviewer #2: Yes

3. Have the authors made all data underlying the findings in their manuscript fully available?

Reviewer #1: Yes

Reviewer #2: Yes

4. Is the manuscript presented in an intelligible fashion and written in standard English?

Reviewer #1: Yes

Reviewer #2: No

5. Review Comments to the Author

Reviewer #1: Overall the paper is good. I have comments that need to be addressed, which are as follows:

1) In abstract, you need to add some results and implications.

2) In introduction, please add specific research question and objectives of research.

3) There is need to add a justification in methodology section and answers why you choose QFD, why not other similar methods? what is the advantages and disadvantages of the methods and how your selected method will help you to address your research objectives.

4) It is highlight recommended to add managerial, practical, and theoretical implications of your research results.

5) Although I am not a native speaker of the language. It is recommended to proof read your paper one more time for consistency, and removing errors.

Reviewer #2: The paper provides an application of quality function deployment (QFD) to improving the resilience of agrifood supply chains. This is a useful QFD case with relatively novel application, that QFD and supply chain practitioners may find valuable in teaching and practice.

There are some minor typographical inaccuracies to be fixed:

A lack of capitalisations following full stops (e.g. lines 100, 123, 271, 297).

(Line 445) reference is made to "Zafar and Haq's research". There is no other mention of this paper in the article, nor is there a numerical reference given for this paper.

Some sentences do not make much logical / rhetorical sense, e.g.

(line 97) "The literature on risk management in agri-food supply chain is extensive but still very limited." How is that possible?

(Line 108) "political and institutional risks, and political and security risks" repetition of political.

(Line 442) "Using IT can effectively reduce the risk of getting stuck in traffic jam" Really? How???

The more significant correction to be made is in the methodology. The sample size (of 341 customers) is justified in the paper with reference to Hair et al and a minimum sample size of 100. But this lacks context: reliable sample size is dependent on the statistical analysis being conducted (e.g. the power of a test, type II risk etc.). At this stage in the paper, however, the proposed statistical analysis is not yet explained. In the minimum number of 100 responses, Hair et al are referring to multivariate methods such as principal component analysis, SEM etc. However, they also describe a minimum number of responses per Likert question on a questionnaire. Other minimum or target sample sizes are cited by Hair et al for Cronbach's alpha reliability values. The authors should explain what they intend to do with their sample data before discussing the sample size or reliability values.

round line 260, the authors discuss Cronbach's alpha "which means that the data is reliable to use". Firstly, data is a pluralisation of the singular datum (so data are....), secondly, again - the authors must explain what they plan to do with the data before discussing C.A. In this case they are planning to produce a summated scale value to measure a construct. This should be explained.

6. PLOS authors have the option to publish the peer review history of their article (what does this mean?). If published, this will include your full peer review and any attached files.

Reviewer #1: No

Reviewer #2: No

---

## [Author Response · Author response to Decision Letter 0]

27 Dec 2021

Response to Reviewer 1

1. Review: In abstract, you need to add some results and implications.

Response: We have added an explanation of the results and implications in the abstract on lines 8 -14.

“The result shows that the top three customer needs are "attractive, bright color", "firm texture" and "fresh smell". The top three risks in the agri-food supply chain are "improper storage," "Harvest Failure" and "Human Resource Risks" and the top three resilience actions are "continuous training," "preventive maintenance," and "supply chain forecasting." The implications of this study are to propose an idea that broadens the perspective of supply chain resilience in the agri-food industry by incorporating the needs of customers in considering how to mitigate existing risks to the satisfaction of customers, and it also highlights the relatively low skill and coordination of the workforce in agri-food supply chains.”

2. Review: In introduction, please add specific research question and objectives of research.

Response: We have added specific explanations of the research questions and objectives in the introduction in lines 47-54. 

“The objectives of this research are as follows. First, to determine the priority of customer needs. Second, to determine the priorities of risks in the agri-food supply chain. Third, to determine the priority of actions to improve supply chain resilience for small and medium-sized agri-food enterprises by identifying customer needs, risks that affect customer satisfaction, and actions that need to be taken to mitigate these risks. We compose the research questions based on the research objectives as follows:

1. What are the priority needs of agri-food customers (CNs)?

2. What are the priority risks in the food supply chain (AFSCRs)?

3. What are the priority supply chain resilience actions (SCRAs)?”

3. Review: There is need to add a justification in methodology section and answers why you choose QFD, why not other similar methods? what is the advantages and disadvantages of the methods and how your selected method will help you to address your research objectives.

Response: We have added a specific explanation in the introduction in lines 182-187 of why we chose QFD, the benefits of choosing the QFD method, and how QFD can achieve our research objectives.

“A model based on the QFD method is used. This method is known for providing an in-depth understanding of customer needs and then used to identify alternative solutions using the House of Quality (HoQ) matrix, which is able to define the relationship between customer needs and products (goods or services). In accordance with the objectives of this study, QFD is an appropriate method to be used in this study because it has the advantage of translating customer needs into attributes ("how") as a form of follow-up by each functional area in meeting customer needs”

4. Review: It is highlight recommended to add managerial, practical, and theoretical implications of your research results.

Response: We have added an explanation of the managerial, practical, and theoretical implications in the implication line in sections 446-466.

 “This result gives rise to several theoretical implications. First, this study is able to provide new ideas and successfully fill the literature gap presented in Section 2 by considering attributes of customer needs and combining them with risk attributes to find resilience solutions that can also satisfy customers in the agri-food supply chain of SMEs. Secondly, this study is able to provide ideas for future researchers as a reference for the practical procedural steps in using two levels of HoQ as a tool to determine the what and how in the QFD method. The thorough understanding gained from a recent literature review on agri-food quality, agri-food risk and supply chain resilience provides the basis for identifying and developing attributes for customer needs, agri-food risk and resilience to build HoQ, which is also confirmed by customers and relevant industry stakeholders. Third, this study proposes an idea for a list of customer needs, risk and resilience solutions that could become a standard for future research. 

This study also has practical implications for agri-food SMEs to build a resilient agri-food supply chain. Firstly, this study has succeeded in highlighting the low skills and knowledge of agri-food actors, especially in the area of supply chain and total quality, in order to raise enthusiasm for improving skills. In addition, agri-food SMEs are weak in coordination among organizations in the supply chain as they are only concerned about internal operational benefits. In view of the findings of this study, it is recommended that agri-food SMEs, starting from producers to suppliers to distributors and retailers, can improve coordination and collaboration to overcome customer satisfaction risks.

In addition, important implications for management also emerge from this study. This industry is a labor-intensive industry. The results show that the priority risks that can negatively affect customer satisfaction are related to the lack of skills and knowledge of employees. Therefore, business owners or top managers should pay more attention to employees' skills and knowledge through continuous training”

5. Review: Although I am not a native speaker of the language. It is recommended to proof read your paper one more time for consistency, and removing errors.

Response: We have proofread this manuscript again and made linguistic improvements. 

We have improved the quality of the English subtitles for this manuscript and have also done some proofreading

Response to Reviewer 2

1. Review: There are some minor typographical inaccuracies to be fixed:

A lack of capitalizations following full stops (e.g. lines 100, 123, 271, 297).

(Line 445) reference is made to "Zafar and Haq's research". There is no other mention of this paper in the article, nor is there a numerical reference given for this paper.

Response: We have corrected some typographical inaccuracies in lines 100, 123, 271 and 297 at your request, which has also changed the line position (please check lines 93, 113, 247 and 272). we have fixed the error by adding a numerical reference to the research of Zafar and Haq (please check line 411)

2. Review: 

a. Some sentences do not make much logical / rhetorical sense, e.g.

(line 97) "The literature on risk management in agri-food supply chain is extensive but still very limited." How is that possible? 

b. (Line 108) "political and institutional risks, and political and security risks" repetition of political. 

c. (Line 442) "Using IT can effectively reduce the risk of getting stuck in traffic jam" Really? How??? 

Response: 

a. We have corrected the sentences to make them more logically/rhetorically meaningful. We use "significant" instead of "extensive" and change it to "The literature on risk management in agri-food supply chain is significant, but still very limited." (Please check line 82).

b. We corrected the word repetition "political" from "political and institutional risk and political and security risk" that you seem to mean in line 100. Change it to "public and institutional political risk and political and security risk" (please check line 93).

c. The use of IT such as software or online mapping applications that can provide information on road density and offer real-time alternative route options with shorter journey times and the best departure times to avoid the risk of congestion.

3. Review: The more significant correction to be made is in the methodology. The sample size (of 341 customers) is justified in the paper with reference to Hair et al and a minimum sample size of 100. But this lacks context: reliable sample size is dependent on the statistical analysis being conducted (eg the power of a test, type II risk etc.). At this stage in the paper, however, the proposed statistical analysis is not yet explained. In the minimum number of 100 responses, Hair et al are referring to multivariate methods such as principal component analysis, SEM etc. However, they also describe a minimum number of responses per Likert question on a questionnaire. Other minimum or target sample sizes are cited by Hair et al for Cronbach's alpha reliability values. The authors should explain what they intend to do with their sample data before discussing the sample size or reliability values.

round line 260, the authors discuss Cronbach's alpha "which means that the data is reliable to use". First, the data is a pluralization of the singular datum (so the data are....), secondly, again - the authors must explain what they plan to do with the data before discussing C.A. In this case they are planning to produce a summated scale value to measure a construct. This should be explained.

Response: The statistical analysis of this study aims to evaluate and present the importance of CNs by calculating the mean value of each CNs attribute. For this purpose, a Likert scale (1-5) questionnaire was used to conduct a survey among 341 customers. 

This study is not a hypothesis testing study. It uses not only the principle of statistical generalization (the sample represents the population), but also that of analytical generalization (generalization from a particular to a broader theory or construct). 

We agree with the 2nd reviewer's opinion about the lack of context in using Hair et al. (2014) as a reference for sample size, and therefore changed the reference for determining our sample size based on the Roscoe rule of thumb, which we cited from Sekaran and Bougie (2016) that stated A sample size of more than 30 and less than 500 is appropriate for most studies. A sample size that is too large (e.g., over 500) could become a problem in that we would then be susceptible to committing Type II errors.

Related Round Line 260 We changed the phrase from data "is" to "are" (please check line 237). We added an explanation of what we intend to do with the data, which is to create a summed scale score to measure a construct, which is done in this study by starting to assess the importance of the CNs by calculating the mean for each CNs attribute. (Please check line 228-230).

We have improved the quality of the English subtitles for this manuscript and have also done some proofreading

---

## [Editor Report · Decision Letter 1]

19 Jan 2022

From resilience to satisfaction: defining supply chain solutions for agri-food SMEs through quality approach

PONE-D-21-28048R1

Dear Dr. Wicaksono,

We’re pleased to inform you that your manuscript has been judged scientifically suitable for publication and will be formally accepted for publication once it meets all outstanding technical requirements.

Kind regards,

Claudio Soregaroli

Academic Editor

PLOS ONE

---

## [Editor Report · Acceptance letter]

21 Jan 2022

PONE-D-21-28048R1 

From resilience to satisfaction: defining supply chain solutions for agri-food SMEs through quality approach 

Dear Dr. Wicaksono:

I'm pleased to inform you that your manuscript has been deemed suitable for publication in PLOS ONE. Congratulations! Your manuscript is now with our production department. 

Kind regards, 

on behalf of

Dr. Claudio Soregaroli 

Academic Editor

PLOS ONE